# Prevalence of intestinal protozoan parasites among school children in africa: A systematic review and meta-analysis

**Khalid Hajissa**[1,2‡], **Md Asiful Islam**[3‡], **Abdoulie M. Sanyang**[4], **Zeehaida Mohamed**[1,5]*

**1** Department of Medical Microbiology & Parasitology, School of Medical Sciences, Universiti Sains Malaysia, Kubang Kerian, Malaysia, **2** Department of Zoology, Faculty of Science and Technology, Omdurman Islamic University, Omdurman, Sudan, **3** Department of Haematology, School of Medical Sciences, Universiti Sains Malaysia, Kubang Kerian, Malaysia, **4** National Public Health Laboratories, Ministry of Health and Social Welfare, Banjul, The Gambia, **5** Medical Microbiology Laboratory, Hospital Universiti Sains Malaysia, Kubang Kerian, Malaysia

‡ These authors share first authorship on this work.
* zeehaida@usm.my

**Data Availability Statement:** All relevant data are within the manuscript and its Supporting Information files.

## Abstract

### Introduction

Parasitic infections, especially intestinal protozoan parasites (IPPs) remain a significant public health issue in Africa, where many conditions favour the transmission and children are the primary victims. This systematic review and meta-analysis was carried out with the objective of assessing the prevalence of IPPs among school children in Africa.

### Methods

Relevant studies published between January 2000 and December 2020 were identified by systematic online search on PubMed, Web of Science, Embase and Scopus databases without language restriction. Pooled prevalence was estimated using a random-effects model. Heterogeneity of studies were assessed using Cochrane Q test and $I^2$ test, while publication bias was evaluated using Egger's test.

### Results

Of the 1,645 articles identified through our searches, 46 cross-sectional studies matched our inclusion criteria, reported data from 29,968 school children of Africa. The pooled prevalence of intestinal protozoan parasites amongst African school children was 25.8% (95% CI: 21.2%-30.3%) with *E. histolytica/ dispar* (13.3%; 95% CI: 10.9%-15.9%) and *Giardia* spp. (12%; 95% CI: 9.8%-14.3%) were the most predominant pathogenic parasites amongst the study participants. While *E. coli* was the most common non-pathogenic protozoa (17.1%; 95% CI: 10.9%-23.2%).

### Conclusions

This study revealed a relatively high prevalence of IPPs in school children, especially in northern and western Africa. Thus, poverty reduction, improvement of sanitation and

**Funding:** The author(s) received no specific funding for this work.

**Competing interests:** The authors have declared that no competing interests exist.

hygiene and attention to preventive control measures will be the key to reducing protozoan parasite transmission.

## Author summary

Pathogenic intestinal protozoan parasites (IPPs) remain a major public health concern. Studies have documented that, the prevalence rates of protozoan infections are quite high in developing regions, particularly Africa and children are the primary victims. Despite numerous studies have been conducted on IPPs in school children in African countries, data on the burden of these infections in African school children have not yet been synthesised. This systematic review and meta-analysis was conducted to provide continent-wide prevalence of IPPs amongst African school children. Our study found that about 25.8% of the children had one or more species of intestinal protozoan parasites in their faecal specimens. *E. histolytica/ dispar* and *Giardia* spp. were the most predominant parasites amongst the study participants. The relatively high prevalence estimate of IPPs amongst African children and the considerable variation in the disease prevalence over the years, between and within countries and regions clearly indicates the needs to improve sanitation and hygiene, paying more attention to preventive control measures as well as poverty reduction which are the key to reducing protozoan parasite transmission.

## Introduction

Despite the significant improvements in health facilities and quality of medical services in terms of diagnosis and mass treatment of parasitic diseases, most of them are still considered major public health problems [1,2]. Infections caused by intestinal protozoan parasites (IPPs) are among the most prevalent human diseases that affect a large section of poor communities particularly in developing countries [3,4]. They have been recognised as significant causes of gastrointestinal illnesses, malnutrition and substantial mortality. Several pathogenic protozoan parasites are responsible for the above health issues including *Entamoeba histolytica/dispar*, *Giardia lamblia* (also known as *Giardia intestinalis* and *Giardia duodenalis*), *Cryptosporidium* and *Balantidium coli*, which are the most common species associated with significant illnesses [3,5,6]. Infection by *E. histolytica* is considered the third most common cause of death after malaria and schistosomiasis [7]. In addtion, *Cryptosporidium* spp. and *G. lamblia* are important nonviral causes of diarrhoeal diseases in humans [8], while other species of intestinal protozoa are either not widely prevalent or non-pathogenic parasites.

Studies have documented that, the prevalence rates of protozoan infections are quite high in developing regions, particularly Africa, and people there are often infected with one or multiple protozoan parasites [9]. The high prevalence of the pathogenic and non-pathogenic protozoa in this continent is intimately related to poverty, poor environmental conditions, lack of access to clean water and adequate sanitation, inadequate hygiene practices and ignorance of health-promoting behaviours [10]. Despite people of all ages are at risk of being infected by intestinal protozoa, children are the most vulnerable and more likely to present with clinical symptom. Furthermore, school children aged 5–17 years are disproportionately affected and often heavily infected because of their habits of playing or handling infested soil, performing unhygienic toilet practices and eating or drinking with soiled hands [11].

Baseline data on the burden, distribution and trend of IPPs can provide essential evidence for implementation of effective prevention strategies in combating these neglected protozoan infections [12]. In this regard, the number of published articles on the epidemiology of IPPs have remarkably increased in recent years. Several studies have been conducted on IPPs in school children in African countries. Hence, there is a need for summarising and critically analysing the available studies to estimate the overall prevalence. To date, no systemic review or meta-analysis on the prevalence of IPPs among school children in Africa has been conducted. Thus, the present study aimed to synthesise existing data on the prevalence of IPPs among school children in Africa, in order to generate much needed, contemporary and reliable continent-wide estimates which might be helpful in the implementation of the relevant prevention and control measures.

## Methods

This study systematically reviewed and analysed the published articles by using the meta-analysis approach to estimate the prevalence of intestinal protozoan parasites among school children in Africa. Literature search, selection of publications and reporting of results were conducted according to the PRISMA guidelines (S1 Checklist) [13]. The protocol of this systematic review and meta-analysis was registered on the International Prospective Register of Systematic Reviews (PROSPERO) database. The registration number is CRD42021233119.

### Search strategy

A comprehensive literature search was performed using all identifed keywords in four electronic databases (PubMed, Web of Science, Embase and Scopus) for the identification of relevant studies that report the prevalence of intestinal protozoan parasites among school children in Africa from January 2000 to December 2020. No language restriction was applied. Moreover, a manual search was conducted using references from retrieved articles for the identification of additional relevant studies that we might have missed. The detailed search strategy for all databases is presented in the S1 Table.

### Data management and study selection

All identified articles were initially retrieved and managed using Endnote X8. After the removal of duplicates, relevant studies were selected independently by two authors (KH and AMS). The titles and abstracts of the retrieved studies were evaluated on the basis of the eligibility criteria. Subsequently, articles with any potential to be eligible for inclusion or any uncertainty about eligibility were further subjected to a full text review. Any disagreement or uncertainty was resolved by discussion, and when necessary, by a third reviewer (MAI). Furthermore, attempts have been made to gain missing data or to clarify any uncertainty with corresponding authors. Articles reporting the same research data/findings published in different formats/titles by the same author were counted only once.

### Inclusion and exclusion criteria

The eligibility of full text articles to be included in this study was evaluated using the following inclusion criteria: (1) cross-sectional studies; (2) conducted in Africa and reporting prevalence of intestinal protozoan parasites; and (3) published between 1 January 2000 and 30 December 2020. The exclusion criteria were as follows: (1) case reports, reviews and studies without original data; (2) non cross-sectional studies; (3) overall prevalence was not reported and impossible to estimate on the basis of the results and confusing or unclear analysis results; (4) survey

was conducted in a hospital or healthcare facilities; (5) and articles that had limited access and those of authors who did not respond to email two times.

### Definition of intestinal protozoan infection and outcome measures

In the context of this study, an IPPs were defined as detection of one or more of the following intestinal parasites: *E. histolytica/dispar*, *Giardia* spp., *Cryptosporidium* spp., *E. coli* and other non-pathogenic protozoan parasites. The main outcome of this systematic review and meta-analysis was the estimated pooled prevalence of IPPs among school children in Africa. The prevalence of IPPs was defined as the proportion of positive samples to the total number of samples.

### Data extraction

Relevant data from each eligible article was extracted and entered into a predefined Excel spreadsheet by the two authors (KH and AMS). Before the inclusion of data in the review, extracted information was checked twice by KH and MAI to ensure consistency and the absence of bias and minimise errors. The following data were extracted: first author's name, year of publication, children enrolment time, country and region where the study was conducted, gender, diagnostic method, sample size, total number of cases, identified species and number of identified species. The United Nations Statistics Division (UNSD) African region (Northern, Eastern, Central, Western and Southern Africa) was assigned to each study according to the country of recruitment.

### Quality assessment

The methodological quality of each included study was appraised by two independent authors (KH and MAI) using the Joana Brigg's Institute (JBI) for prevalence studies [14], having nine checklist items with four options: 'yes', 'no', 'unclear' and 'not applicable' were used. The final score of each article was calculated according to the proportion of 'yes' answers. Studies were categorised as 'high risk of bias' (low quality), 'moderate risk of bias' (moderate quality) or 'low risk of bias' (high quality) when the overall score was $\leq$ 49%, 50%–69% or $\geq$70% respectively [15,16].

### Data analysis

The prevalence estimate and corresponding 95% confidence interval (CI) were calculated for each included study. The prevalence data were then pooled through statistical meta-analysis with the restricted maximum likelihood (REML) method for random-effects model. A forest plot was generated to present the summarised results and heterogeneity among the included studies. Heterogeneity among studies was assessed using $I^2$ statistics, in which $I^2$ values of greater than 75% inidicated substantial heterogeneity [17]. The significance of heterogeneity was identified using Cochran's Q-test. Publication bias was checked visually using a funnel plot and objectively using Egger's regression test.

The potential sources of heterogeneity were further explored by subgroup analysis according to children enrolment time, detection method, region and sample size. Furthermore, the robustness of the pooled estimate was tested through sensitivity analysis according to the following strategies: (i) excluding small studies (n < 200); (ii) excluding moderate-quality studies (moderate risk of bias); (iii) excluding studies that used non-microscopic detection methods; and (iv) excluding outlier studies. Data analysis was performed and a plot was created using

metaprop codes in the meta (version 4.15–1) and metafor (version 2.4–0) packages of R (version 3.6.3) in RStudio (version 1.3.1093).

## Results

### Study selection

A total of 1,645 articles were initially identified form the four databases. After 707 duplicates were removed, another 785 studies were excluded from the remaining articles after title and/or abstract evaluation. Furthermore, 107 articles were further excluded during the full text assessment with reasons (S2 Table). Finally, only 46 (2.8%) of the articles met the eligibility criteria and included in the systematic review and meta-analysis (Fig 1).

### Characteristics of included studies

The detailed characteristics of the included studies are summarised in Table 1. The 46 eligible studies were conducted in 19 African countries, across the five UNSD regions of Africa. Ethiopia had the highest number of eligible studies (17 studies), followed by Nigeria (six studies) and

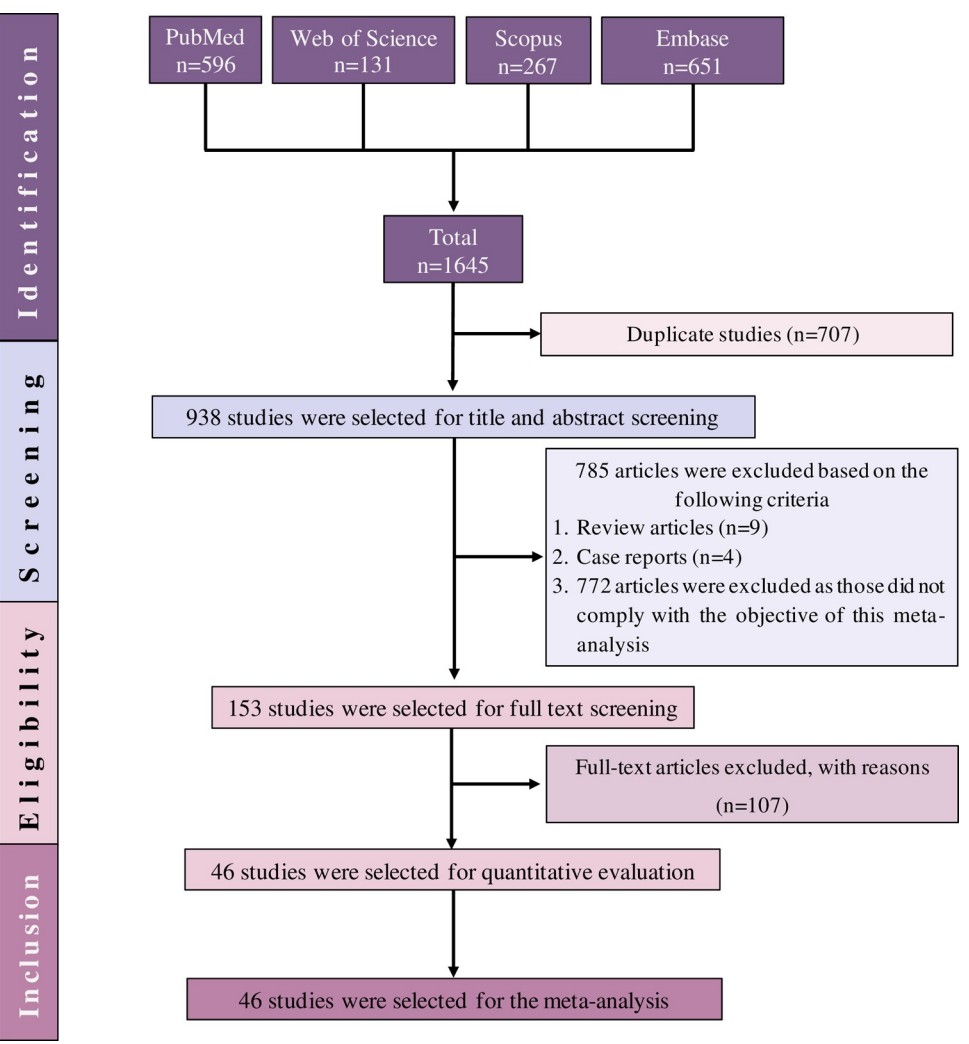

**Fig 1. PRISMA flow diagram of study selection.**

**Table 1. Major characteristics of the included studies.**

| No | Study ID (references) | Publication Year | Country, place | Sample size (% female) | Cases | Methods | Reported parasites |
|---|---|---|---|---|---|---|---|
| 1 | Abdel-Aziz 2010 [18] | 2010 | Sudan, El dhayga, Central Sudan | 157 (47.8) | 83 | DWM and FECT | *E. histolytica* and *G. lamblia* |
| 2 | Abossie 2014 [19] | 2014 | Ethiopia, Chencha town, Southern Ethiopia | 400 (52.3) | 94 | DWM and FECT | *E. histolytica/dispar* and *G. lamblia* |
| 3 | Adams 2005 [20] | 2005 | South Africa, Cape Town | 3890 (49.8) | 673 | FECT | *Giardia* spp. |
| 4 | Adedoja 2015 [21] | 2015 | Nigeria, Pategi, Kwara State | 748 (50.8) | 197 | DWM and FECT | *E. histolytica*, *E. coli*, *G. lamblia* |
| 5 | Alemu 2019a [22] | 2019 | Ethiopia, Birbir, Southern Ethiopia | 351 (48.7) | 25 | DWM and FECT | E. histolytica/dispar and G. lamblia |
| 6 | Alemu 2019b [23] | 2019 | Ethiopia, Northwest | 273 (45.8) | 22 | DWM and FECT | *E. histolytica/dispar* and *G. lamblia* |
| 7 | Al-Shehri 2019 [24] | 2019 | Uganda, Gondar town | 254 (50.4) | 221 | qPCR | *G. duodenalis* |
| 8 | Amare 2013 [25] | 2013 | Ethiopia, Gondar town, Northwest | 405 (46.2) | 2 | DWM and FECT | *G. lamblia* and *Entamoeba* spp. |
| 9 | Awolaju 2009 [26] | 2009 | Nigeria, South-west | 312 (54.5) | 29 | DWM | *E. histolytica* |
| 10 | Ayogu 2015 [27] | 2015 | Nigeria, Enugu State | 450 (51.6) | 190 | DWM | *E.histolytica* |
| 11 | Baba 2012 [28] | 2012 | Mauritania, Gorgol, Guidimagha and Brakna | 1308 (57.8) | 405 | DWM | *E. histolytica*, *E. coli*, *E. hartmani*, *G. intestinalis*, *E. nanus*, *Pseudolimax butchilii* and *C. mesnilii* |
| 12 | Birhanu 2018 [29] | 2018 | Ethiopia, Pawe, Northwest Ethiopia | 422 (54.0) | 20 | DWM and FECT | *G. lamblia* |
| 13 | Bisangamo 2017 [30] | 2017 | Kiliba city, Eastern DR Congo | 602 (55.1) | 181 | DWM | *E. histolytica*, *G. intestinalis* and *Trichomonas intestinalis* |
| 14 | Chege 2020 [31] | 2020 | Kenya, Nakuru town | 96 (NR) | 40 | PCR | *E. dispar*, *E. coli* and *G. intestinalis* |
| 15 | de Alegria 2017 [32] | 2017 | Angola, Cubal, Southwestern | 230 (56.1) | 17 | FECT | *G. lamblia*, *B. coli* and *E. histolytica/dispar* |
| 16 | Dyab 2016 [33] | 2016 | Egypt, Aswan | 300 (43.3) | 58 | DWM, FECT and mZN stain | *E. histolytica*, *G. lamblia* and *Cryptosporidium* spp. |
| 17 | Erismann 2016 [34] | 2016 | Burkina Faso, Plateau Central and Centre-Ouest regions | 385 (48.8) | 326 | DWM and FECT | *E. histolytica/dispar*, *E. coli*, *Trichomonas intestinalis*, *B. coli* and *G. intestinalis* |
| 18 | Eyamo 2019 [35] | 2019 | Ethiopia, Tula Sub-City, Southern | 384 (51.8) | 82 | FECT | *G. duodenalis* and *E. histolytica/dispar* |
| 19 | Fan 2012 [36] | 2012 | Kingdom of Swaziland, northwestern | 267 (56.9) | 86 | MIFC | *G. lamblia*, *E. histolytica/dispar*, *B. hominis*, *E. coli*, *E. nana*, *C. mesnelli* and *Iodamoeba butschlii* |
| 20 | Forson 2017 [37] | 2017 | Ghana, Accra | 300 (48.0) | 33 | DWM and FECT | *G. lamblia* and *E. histolytica/dispar* |
| 21 | Gebretsadik 2020 [38] | 2020 | Ethiopia, Harbu, North East | 400 (62.3) | 37 | DWM and FECT | *E. histolytica* and *G. lamblia* |
| 22 | Gelaw 2013 [39] | 2013 | Ethiopia, Gondar town | 304 (44.1) | 40 | DWM and FECT | E. histolytica/dispar and G. intestinalis |
| 23 | Gyang 2019 [40] | 2019 | Nigeria, Lagos city | 384 (51.0) | 202 | MIFC | *E. histolytica/dispar*, *E. coli*, *G. duodenalis*, *E. nana* and *B. hominis* |
| 24 | Hailegebriel 2017 [41] | 2017 | Ethiopia, Bahir Dar | 359 (50.7) | 134 | FECT | *E. histolytica*, *G. lamblia* and *T. hominins* |
| 25 | Hailegebriel 2018 [42] | 2018 | Ethiopia, Bahir Dar | 382 (49.5) | 66 | FECT | *E. histolytica/dispar*, *G. lamblia* and *Isospora belli* |
| 26 | Hall 2008 [43] | 2008 | Ethiopia, all 11 regions of Ethiopia | 7466 (50.2) | 239 | FECT | *G. duodenalis* |
| 27 | Heimer 2015 [44] | 2015 | Rwanda, Huye district | 622 (NR) | 222 | qPCR | *G. duodenalis* |

(*Continued*)

**Table 1.** (Continued)

| No | Study ID (references) | Publication Year | Country, place | Sample size (% female) | Cases | Methods | Reported parasites |
|---|---|---|---|---|---|---|---|
| 28 | Htun 2018 [45] | 2018 | South Africa, Port Elizabeth, South-eastern | 842 (49.4) | 114 | RDTs | *C. parvum* and *G. intestinalis* |
| 29 | Ibrahium 2011 [46] | 2011 | Egypt, Minia Governorate | 264 (64.8) | 84 | DWM and FECT | *G. lamblia* and *E. coli* |
| 30 | Ihejirika 2019 [47] | 2019 | Nigeria, Imo State, South Eastern | 300 (NR) | 32 | FECT | *E. histolytica, E. coli* and *G. lambia* |
| 31 | Jejaw 2015 [48] | 2015 | Ethiopia, Mizan-Aman, Southwest | 460 (50.4) | 36 | DWM and FECT | *G. lamblia* and *E. histolytica/dispar/moshkovskii* |
| 32 | Kesete 2020 [49] | 2020 | Eritrea, Ghindae town | 450 (52.2) | 172 | FECT | *E. histolytica/dispar* and *G. duodenalis* |
| 33 | Khaled 2020 [50] | 2020 | Senegal, northwestern | 731 (48.2) | 588 | qPCR | *Blastocystis* spp. |
| 34 | Legesse 2010 [51] | 2010 | Ethiopia, Adwa, Northern | 381 (56.2) | 7 | FECT | *E. histolytica/dispar* |
| 35 | Liao 2016 [52] | 2016 | DR of Sao Tome and Principe, Capital areas | 252 (52.0) | 72 | MIFC | *E. histolytica/dispar, G. intestinalis* and *B. hominis* |
| 36 | Mahmud 2013 [53] | 2013 | Ethiopia, Northern | 583 (53.5) | 286 | DWM and FECT | *E. histolytica/dispar* and *G. lamblia* |
| 37 | Müller 2016 [54] | 2016 | South Africa, Port Elizabeth | 934 (49.5) | 144 | RDTs | *C. parvums* and *G. intestinalis* |
| 38 | Nguyen 2012 [55] | 2012 | Ethiopia, Angolela | 664 (48.6) | 202 | FECT | *G. lamblia* and *E. histolytica* |
| 39 | Njambi 2020 [56] | 2020 | Kenya, Mwea, Central | 180 (50.0) | 59 | DWM | *E. histolytica/dispar, E. coli* and *G. intestinalis* |
| 40 | Oliveira 2015 [57] | 2015 | Angola, Lubango city, Huíla Province | 328 (56.4) | 66 | DWM | *G. lamblia* and *E. histolytica/dispar* |
| 41 | Opara 2012 [58] | 2012 | Nigeria, Akwa Ibom State | 405 (49.4) | 21 | DWM and FECT | *G. lamblia* and *E. histolytica/dispar* |
| 42 | Orish 2019 [59] | 2019 | Ghana, Volta Region | 550 (54.7) | 11 | DWM | *Entamoeba* spp. |
| 43 | Reji 2011 [60] | 2011 | Ethiopia, Adama town | 358 (57.3) | 57 | FECT | *E. histolytica/dispar* and *G. lamblia* |
| 44 | Sitotaw 2020 [61] | 2020 | Ethiopia, Sasiga District, Southwest | 383 (48.8) | 58 | DWM and FECT | *E. histolytica* and *G. intestinalis* |
| 45 | Tagajdid 2012 [62] | 2012 | Morocco, Salé city | 123 (NR) | 71 | MIFC | *E. histolytica/dispar, G. intestinalis, E. nana, C. mesnilii* and *B. hominis* |
| 46 | Tembo 2020 [63] | 2020 | Zambia, Lusaka | 329 (55.6) | 33 | DWM and FECT | *G. duodenalis* |

NR; not recorded, DWM; Direct wet mount, FECT: formalin-ether concentration technique, MIFC: Merthiolate-iodine-formaldehyde concentration, mZN stain; PCR: Polymerase chain reaction, qPCR: Real-time PCR, RDTs: Rapid Diagnostic Tests

South Africa (three studies). Two studies were conducted in each of the four countries, namely: Angola, Ghana, Kenya and Egypt, and one study was performed in each of the following countries: Burkina Faso, Democratic Republic of the Congo, DR of Sao Tome and Principe, Eritrea, Kingdom of Swaziland, Mauritania, Morocco, Rwanda, Senegal, Sudan, Uganda and Zambia. The included studies were school-based surveys and had cross-sectional study designs. A total of 29,968 school children aged 6–17 years were examined for the presence of IPPs. Microscopy was the predominant detection method for IPPs laboratory confirmation (40 studies). Molecular detection was used in four studies, similar to rapid diagnostic test. A range of protozoan parasites were detected across the studies, including: *Entamoeba histolytica/ dispar, Giardia* spp., *Cryptosporidium* spp., *Entamoeba coli, Entoamoeba hartmanii, Cyclospora cayetanensis, Blastocystis hominis, Endolimax nana* and *Iodamoeba butschli*. A map with the geographical distributions across the continent based on the studies included is presented in Fig 2.

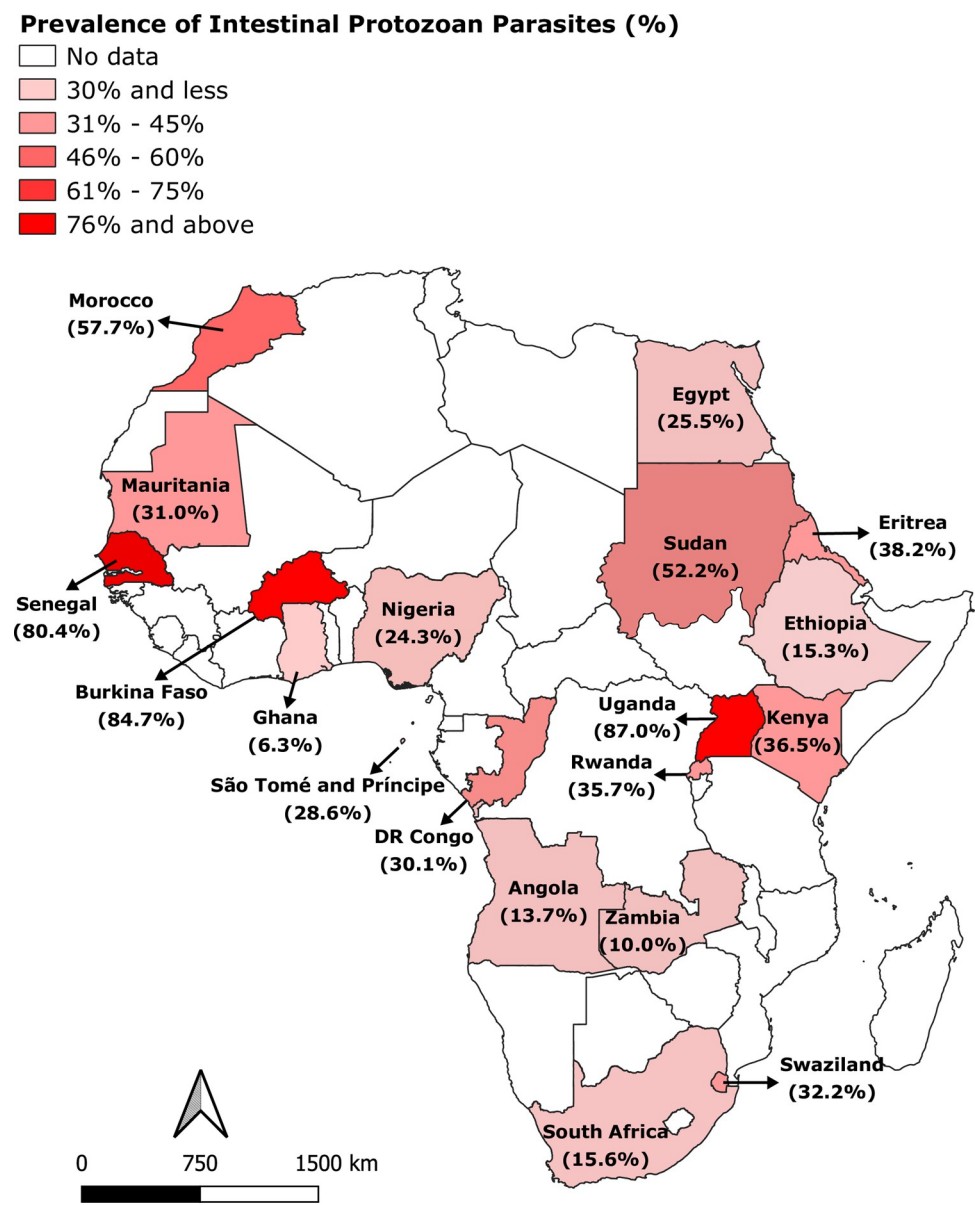

**Fig 2. Prevalence of intestinal protozoan parasites among school children in Africa.** Figure created by authors using QGIS software. Basemap source: https://www.diva-gis.org/Data.

## Pooled prevalence of intestinal protozoan

The prevalence of IPPs among school children in Africa ranged from 0.5% (95% CI: 0.0%-1.2%) in Ethiopia to 87% (95% CI: 82.9%-91.1%) in Uganda (24, 25). An overall prevalence of 25.8% (95% CI: 21.2%-30.3%) was obtained from 7731 school children infected with one or more species of IPPs. Substantial heterogeneity were seen across all the included studies ($I^2$ = 100%, P < 0.001) (Fig 3).

## Quality assessment and publication bias

Information about the individual study quality assessment is presented in S3 Table. Briefly, 58.7% of the included studies were of high quality (low risk of bias), whereas the remaining

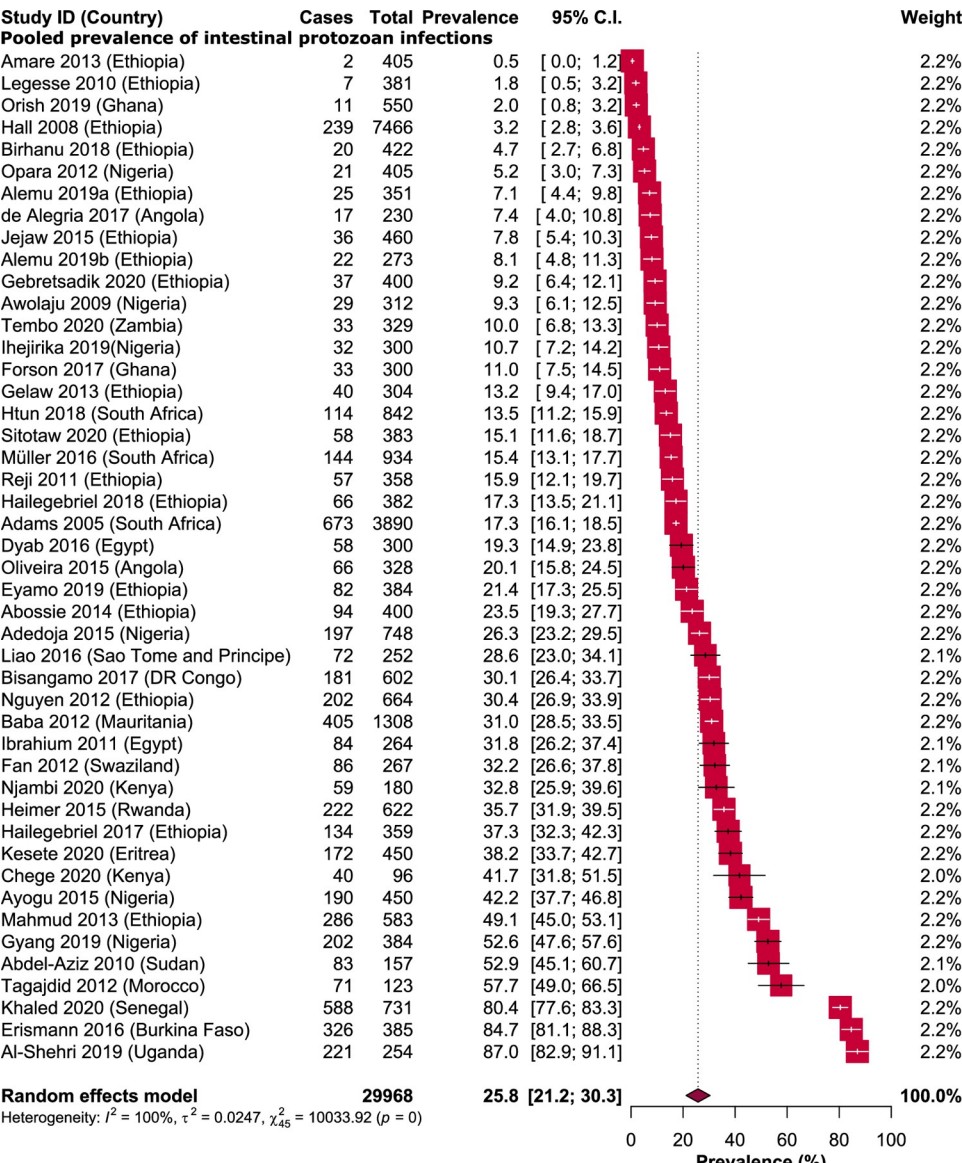

| Study ID (Country) | Cases | Total | Prevalence | 95% C.I. | Weight |
|---|---|---|---|---|---|
| **Pooled prevalence of intestinal protozoan infections** | | | | | |
| Amare 2013 (Ethiopia) | 2 | 405 | 0.5 | [ 0.0; 1.2] | 2.2% |
| Legesse 2010 (Ethiopia) | 7 | 381 | 1.8 | [ 0.5; 3.2] | 2.2% |
| Orish 2019 (Ghana) | 11 | 550 | 2.0 | [ 0.8; 3.2] | 2.2% |
| Hall 2008 (Ethiopia) | 239 | 7466 | 3.2 | [ 2.8; 3.6] | 2.2% |
| Birhanu 2018 (Ethiopia) | 20 | 422 | 4.7 | [ 2.7; 6.8] | 2.2% |
| Opara 2012 (Nigeria) | 21 | 405 | 5.2 | [ 3.0; 7.3] | 2.2% |
| Alemu 2019a (Ethiopia) | 25 | 351 | 7.1 | [ 4.4; 9.8] | 2.2% |
| de Alegria 2017 (Angola) | 17 | 230 | 7.4 | [ 4.0; 10.8] | 2.2% |
| Jejaw 2015 (Ethiopia) | 36 | 460 | 7.8 | [ 5.4; 10.3] | 2.2% |
| Alemu 2019b (Ethiopia) | 22 | 273 | 8.1 | [ 4.8; 11.3] | 2.2% |
| Gebretsadik 2020 (Ethiopia) | 37 | 400 | 9.2 | [ 6.4; 12.1] | 2.2% |
| Awolaju 2009 (Nigeria) | 29 | 312 | 9.3 | [ 6.1; 12.5] | 2.2% |
| Tembo 2020 (Zambia) | 33 | 329 | 10.0 | [ 6.8; 13.3] | 2.2% |
| Ihejirika 2019(Nigeria) | 32 | 300 | 10.7 | [ 7.2; 14.2] | 2.2% |
| Forson 2017 (Ghana) | 33 | 300 | 11.0 | [ 7.5; 14.5] | 2.2% |
| Gelaw 2013 (Ethiopia) | 40 | 304 | 13.2 | [ 9.4; 17.0] | 2.2% |
| Htun 2018 (South Africa) | 114 | 842 | 13.5 | [11.2; 15.9] | 2.2% |
| Sitotaw 2020 (Ethiopia) | 58 | 383 | 15.1 | [11.6; 18.7] | 2.2% |
| Müller 2016 (South Africa) | 144 | 934 | 15.4 | [13.1; 17.7] | 2.2% |
| Reji 2011 (Ethiopia) | 57 | 358 | 15.9 | [12.1; 19.7] | 2.2% |
| Hailegebriel 2018 (Ethiopia) | 66 | 382 | 17.3 | [13.5; 21.1] | 2.2% |
| Adams 2005 (South Africa) | 673 | 3890 | 17.3 | [16.1; 18.5] | 2.2% |
| Dyab 2016 (Egypt) | 58 | 300 | 19.3 | [14.9; 23.8] | 2.2% |
| Oliveira 2015 (Angola) | 66 | 328 | 20.1 | [15.8; 24.5] | 2.2% |
| Eyamo 2019 (Ethiopia) | 82 | 384 | 21.4 | [17.3; 25.5] | 2.2% |
| Abossie 2014 (Ethiopia) | 94 | 400 | 23.5 | [19.3; 27.7] | 2.2% |
| Adedoja 2015 (Nigeria) | 197 | 748 | 26.3 | [23.2; 29.5] | 2.2% |
| Liao 2016 (Sao Tome and Principe) | 72 | 252 | 28.6 | [23.0; 34.1] | 2.1% |
| Bisangamo 2017 (DR Congo) | 181 | 602 | 30.1 | [26.4; 33.7] | 2.2% |
| Nguyen 2012 (Ethiopia) | 202 | 664 | 30.4 | [26.9; 33.9] | 2.2% |
| Baba 2012 (Mauritania) | 405 | 1308 | 31.0 | [28.5; 33.5] | 2.2% |
| Ibrahium 2011 (Egypt) | 84 | 264 | 31.8 | [26.2; 37.4] | 2.1% |
| Fan 2012 (Swaziland) | 86 | 267 | 32.2 | [26.6; 37.8] | 2.1% |
| Njambi 2020 (Kenya) | 59 | 180 | 32.8 | [25.9; 39.6] | 2.1% |
| Heimer 2015 (Rwanda) | 222 | 622 | 35.7 | [31.9; 39.5] | 2.2% |
| Hailegebriel 2017 (Ethiopia) | 134 | 359 | 37.3 | [32.3; 42.3] | 2.2% |
| Kesete 2020 (Eritrea) | 172 | 450 | 38.2 | [33.7; 42.7] | 2.2% |
| Chege 2020 (Kenya) | 40 | 96 | 41.7 | [31.8; 51.5] | 2.0% |
| Ayogu 2015 (Nigeria) | 190 | 450 | 42.2 | [37.7; 46.8] | 2.2% |
| Mahmud 2013 (Ethiopia) | 286 | 583 | 49.1 | [45.0; 53.1] | 2.2% |
| Gyang 2019 (Nigeria) | 202 | 384 | 52.6 | [47.6; 57.6] | 2.2% |
| Abdel-Aziz 2010 (Sudan) | 83 | 157 | 52.9 | [45.1; 60.7] | 2.1% |
| Tagajdid 2012 (Morocco) | 71 | 123 | 57.7 | [49.0; 66.5] | 2.0% |
| Khaled 2020 (Senegal) | 588 | 731 | 80.4 | [77.6; 83.3] | 2.2% |
| Erismann 2016 (Burkina Faso) | 326 | 385 | 84.7 | [81.1; 88.3] | 2.2% |
| Al-Shehri 2019 (Uganda) | 221 | 254 | 87.0 | [82.9; 91.1] | 2.2% |
| **Random effects model** | | 29968 | **25.8** | **[21.2; 30.3]** | **100.0%** |

Heterogeneity: $I^2 = 100\%$, $\tau^2 = 0.0247$, $\chi^2_{45} = 10033.92$ ($p = 0$)

Prevalence (%): 0  20  40  60  80  100

**Fig 3. Forest plot representing the pooled prevalence of intestinal protozoan infections among school children in Africa.**

41.3% were of moderate quality. Funnel plot asymmetry indicated the existence of publication bias among the included studies (Fig 4). Similarly, regression-based Egger's test revealed statistically significant publication bias ($P = 0.001$).

## Subgroup analysis

With evidence of the substantial heterogeneity, subgroup analysis was performed. The results are shown in Table 2 and S1 Fig. According to children enrolment time, prevalence data were pooled into three-year periods for comparison. The prevalence of IPPs was gradually increased from 19.4% during the period between 2005 and 2010 to 23.5% in the next five years (2011–2015), and to 25.2% from 2016 to 2020. Among the UNSD African regions, Northern Africa had the highest prevalence (42.2%; CI: 22.7%-57.6%), followed by Western Africa (32.3%; 95%

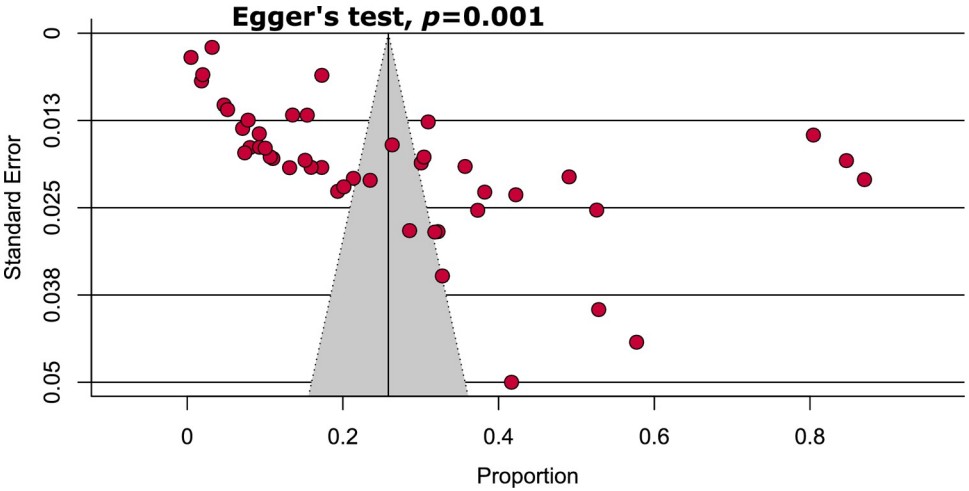

**Fig 4. Funnel plot representing evidence of publication bias**

CI: 15.1%-49.5%), Eastern Africa (21.9%; 95% CI: 17.0% -26.8%) and Central Africa (21.5%; 95% CI: 10.1%-32.8%). Southern Africa had the lowest prevalence of 18.6 (95% CI: 14.5%-22.8%). Notably, remarkable differences in IPPs estimates obtained with laboratory detection methods were observed. A remarkably high overall estimate was observed when PCR or qPCR were used (61.4%; CI: 35.3–87.4%), and the pooled prevalence rates obtained through microscopy or RDTs were 22.7 (95% CI: 18.8–26.6%) or 14.5 (95% CI: 12.6–16.3%), respectively.

## Common intestinal protozoan infections among school children

Of the 46 included studies, *Giardia* spp. (38/46 [82.6%]) and *E. histolytica/ dispar* (33/46 [71.7%]) had the highest number of reports (Table 2). Similarly, *E. histolytica/ dispar* was the most common pathogenic protozoan parasite detected in children (13.3%; 95% CI: 10.9%-15.9%), followed by *Giardia* spp. (12%; 95% CI: 9.8%-14.3%) and *Cryptosporidium* spp. (2.5%; 95% CI: 1.8%-3.2%). Of the non-pathogenic protozoa, *E. coli* was the most common, with a prevalence of 17.1% (95% CI: 10.9%-23.2%).

## Sensitivity analysis

Sensitivity analyses indicated that the exclusion of small studies, studies that used non-microscopic detection methods and outlier studies (Fig 5) did not significantly altered the summary of the pooled estimates. Prevalence rate remained within the 95% CI of the respective overall prevalence (Table 3 and S2 Fig). Despite that the removal of moderate-quality studies reduced the overall prevalence by 9.4%, it did not significantly reduce heterogeneity. Overall, the stability of IPPs prevalence validated the reliability and rationality of our analyses.

## Discussion

Intestinal protozoan infections significantly contribute to the burden of gastrointestinal illnesses throughout Africa, where many conditions favour the transmission and children are the primary victims [64,65]. Here, we present the first systematic review and meta-analysis of the continent-wide prevalence of IPPs amongst school children. The current review compiled eligible data on the prevalence of IPPs from 29,968 school children reported in 46 studies conducted in 19 African countries. The prevalence rates of IPPs in African school children varied

**Table 2. Pooled prevalence of intestinal protozoan infections in different subgroups.**

| Subgroups | Prevalence [95% CIs] (%) | Number of studies analysed | Total number of subjects | Heterogeneity $I^2$ | Heterogeneity $p$-value | Publication Bias, Egger's test ($p$-value) |
|---|---|---|---|---|---|---|
| *Children enrolment time* | | | | | | |
| Year 2005–2010 | 19.4 [12.5–26.4] | 9 | 3,168 | 99% | <0.0001 | NA |
| Year 2011–2015 | 23.5 [9.3–37.7] | 10 | 4,107 | 99% | <0.0001 | 0.45 |
| Year 2016–2020 | 25.2 [6.9–43.4] | 9 | 3,314 | 100% | <0.0001 | NA |
| *Different regions of Africa* | | | | | | |
| Northern Africa | 40.2 [22.7–57.6] | 4 | 844 | 97% | <0.0001 | NA |
| Eastern Africa | 21.9 [17.0–26.8] | 23 | 15,906 | 99% | <0.0001 | 0.02 |
| Central Africa | 21.5 [10.1–32.8] | 4 | 1,412 | 97% | <0.0001 | NA |
| Western Africa | 32.3 [15.1–49.5] | 11 | 5,873 | 100% | <0.0001 | 0.20 |
| Southern Africa | 18.6 [14.5–22.8] | 4 | 5,933 | 92% | <0.0001 | NA |
| *Countries* | | | | | | |
| Ethiopia | 15.3 [11.7–19.0] | 17 | 13,975 | 99% | <0.0001 | <0.0001 |
| Nigeria | 24.3 [10.7–37.8] | 6 | 2,599 | 99% | <0.0001 | NA |
| South Africa | 15.6 [13.3–17.9] | 3 | 5,666 | 77% | 0.01 | NA |
| Angola | 13.7 [1.2–26.2] | 2 | 558 | 95% | <0.0001 | NA |
| Ghana | 6.3 [0.0–15.2] | 2 | 850 | 96% | <0.0001 | NA |
| Kenya | 36. 5 [27.9–45.1] | 2 | 276 | 52% | 0.14 | NA |
| Egypt | 25.5 [13.2–37.7] | 2 | 564 | 91% | <0.0001 | NA |
| Burkina Faso | 84.7 [81.1–88.3] | 1 | 385 | NA | NA | NA |
| DR Congo | 30.1 [26.4–33.7] | 1 | 602 | NA | NA | NA |
| Eritrea | 38.2 [33.7–42.7] | 1 | 450 | NA | NA | NA |
| Mauritania | 31.0 [28.5–33.5] | 1 | 1308 | NA | NA | NA |
| Morocco | 57.7 [49.0–66.5] | 1 | 123 | NA | NA | NA |
| Rwanda | 35.7 [31.9–39.5] | 1 | 622 | NA | NA | NA |
| Sao Tome Principe | 28.6 [23.0–34.1] | 1 | 252 | NA | NA | NA |
| Senegal | 80.4 [77.6–83.3] | 1 | 731 | NA | NA | NA |
| Sudan | 52.9 [45.1–60.7] | 1 | 157 | NA | NA | NA |
| Swaziland | 32.2 [26.6–37.8] | 1 | 267 | NA | NA | NA |
| Uganda | 87.0 [82.9–91.1] | 1 | 254 | NA | NA | NA |
| Zambia | 10.0 [6.8–13.3] | 1 | 329 | NA | NA | NA |
| *Different diagnostic methods* | | | | | | |
| Microscopy | 22.7 [18.8–26.6] | 40 | 26,489 | 99% | <0.0001 | <0.0001 |
| PCR or qPCR | 61.4 [35.3–87.4] | 4 | 1,703 | 99% | <0.0001 | NA |
| Rapid diagnostic kit | 14.5 [12.6–16.3] | 4 | 1,776 | 21% | 0.16 | NA |
| *Different species* | | | | | | |
| *Giardia* spp. | 12.0 [9.8–14.3] | 38 | 26,565 | 99% | <0.0001 | 0.0003 |
| *E. histolytica/ dispar* | 13.3 [10.7–15.9] | 33 | 13,235 | 99% | <0.0001 | <0.0001 |
| *Entamoeba coli* | 17.1 [10.9–23.2] | 9 | 3,788 | 97% | <0.0001 | NA |
| *Cryptosporidium* spp. | 2.5 [1.8–3.2] | 3 | 2,076 | 3% | 0.35 | NA |

CIs: Confidence intervals; NA: Not applicable.

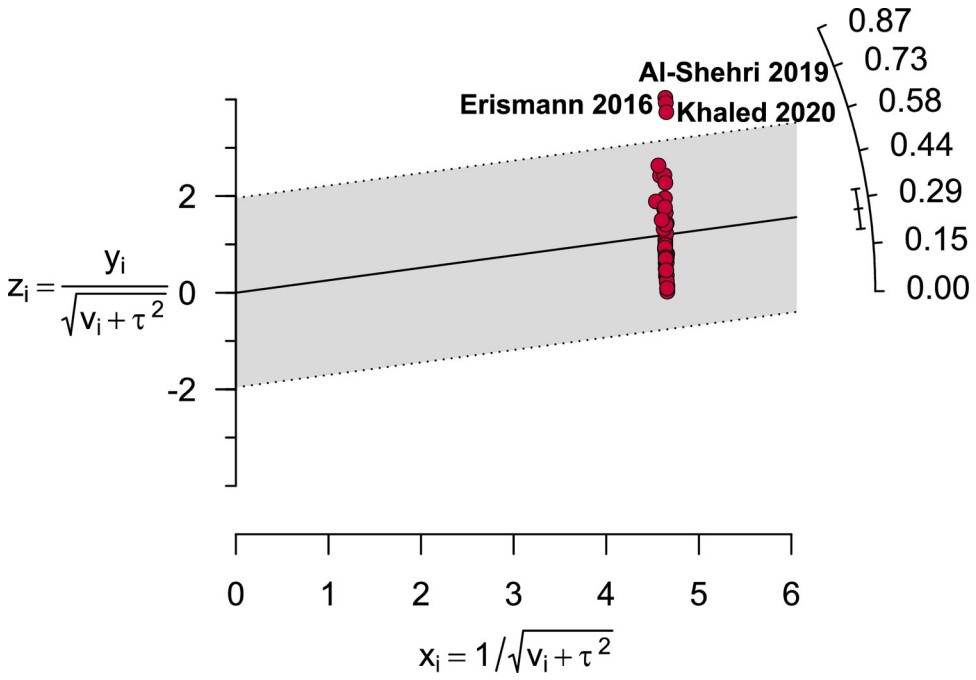

$$z_i = \frac{y_i}{\sqrt{v_i + \tau^2}}$$

$$x_i = 1/\sqrt{v_i + \tau^2}$$

**Fig 5. Galbraith plot depicting three outlier studies.**

greatly amongst the included studies. According to Fig 2, the highest and lowest prevalence rates of IPPs were reported in studies conducted in Uganda (87%, 95% CI: 82.9%-91.1%) [24] and Ethiopia (0.5%, 95% CI: 0.0%-1.2%), respectively [25]. Such considerable variation is not surprising given that environmental conditions and socioeconomic status vary between and within the countries and different detection methods are used. In this review, the prevalence of IPPs amongst children was 25.8% (95% CI: 21.2%-30.3%), which could be due to poor hygiene given that the disease is transmitted via food, water and fingers that are contaminated with faeces. The relatively high number (7,731) of school children with IPPs in Africa in the present study is aligned with the 24.2% infection rate reported in Thailand [66]. However, our finding is higher than the data in Iran (16.9%) [67]. The difference might be attributed to the aforementioned reasons in addition to personal and cultural habits.

Significant decreasing trends of IPPs were observed amongst children in Nepal [68] and India [69], which could be due to improvement in sanitation and hygiene, socioeconomic

**Table 3. Sensitivity analyses.**

| Strategies of Sensitivity analyses | Prevalence [95% CIs] (%) | Difference of pooled prevalence compared to the main result | Number of studies analysed | Total number subjects | Heterogeneity | |
|---|---|---|---|---|---|---|
| | | | | | $I^2$ | $p$-value |
| Excluding small studies | 23.9 [19.2–28.6] | 1.9% lower (2.0% lower—1.7% lower) | 42 | 29,412 | 100% | <0.0001 |
| Excluding moderate-quality studies | 16.4 [12.9–19.8] | 9.4% lower (8.3% lower—10.5% lower) | 27 | 21,064 | 99% | <0.0001 |
| Excluding studies used non-microscopic detection methods | 22.7 [18.8–26.6] | 3.1% lower (2.4% lower—3.7% lower) | 40 | 26,489 | 99% | <0.0001 |
| Excluding outlier studies | 21.4 [18.2–24.6] | 4.4% lower (3.0% lower—5.7% lower) | 43 | 28,598 | 99% | <0.0001 |

CIs: confidence intervals

development and establishing preventive control measures and control strategy. By contrast, the present findings revealed that the magnitude of IPPs gradually increased from 19.4% in 2005–2010 to 23.5% and 25.2% in 2011–2015 and 2016–2020, repectively. The increasing trend could be attributed to insufficient financial support, lack of political commitment and inadequate community involvement in implementation of effective strategies to reduce the infection in Africa [70].

The findings of this systematic review and meta-analysis indicated that northern and western Africa had the highest prevalence estimates (42.2% and 32.3%) than eastern (21.9%), central (21.5%) and southern Africa (18.6%). Whether exposure to IPPs through poor hygiene is higher in northern and western Africa than in other parts of the continent or/and whether it is related to environmental condition remains unknown. High prevalence of IPPs was also reported in eastern, central and southern Africa. Therefore, comparison of overall prevalence rates by regions may not provide sufficient detailed information, and additional studies are needed to further explore the sources of variation.

Africa consists of 54 countries, but IPPs was only reported in 19 countries. Ethiopia had the highest number of eligible studies (17 studies), with overall prevalence rate of 15.3%, which is lower than the 24.21% rate reported in 2020 by Tegen *et al.* [4] in the same geographical area. The second and third highest numbers of studies included were from Nigeria (six studies) and South Africa (three studies), respectively. Only two studies were reported in Angola, Ghana, Kenya and Egypt. Although the outcome of one study is inconclusive and cannot be generalised, only one eligible study was identified in the 12 remaining countries. Moreover, data from studies in 35 countries were unavailable because they did not met our eligibility criteria. Hence, further studies with different inclusion/exclusion criteria are needed, and scholars should focus on IPPs amongst school children in these countries.

Different parasitological techniques are used because of lack of gold standard test (with 100% accuracy) for detection of intestinal parasites. The prevalence estimate obtained by microscopy was lower (22.7%) than that achieved when using molecular methods (61.4%) but slightly higher than when using RDTs (14.5%). The differences in laboratory techniques used for IPPs diagnosis and the variations in the sensitivity and specificity even of same method could possibly be the reason for the observed disparity in the IPPs rates in the present study. The use of DNA-based methods for laboratory confirmation of intestinal parasites has been proven to be highly sensitive and specific [71,72]. This finding is evidenced by the significantly high prevalence (61.4%) of IPPs in the present study when PCR or qPCR was used. However, such methods require specialised equipment and technical expertise of personnel, which limit their use. As such, traditional stool examination (microscopy) is still widely used for diagnosis of protozoan parasites worldwide [9,73]. About 87% (40/46) of the included studies used microscopy as detection methods.

In this meta-analysis, nine types of protozoan parasites were identified; *Giardia* spp. (38/46 [82.6%]) and *E. histolytica/ dispar* (33/46 [71.7%]) were the most frequently reported parasites (Table 2). The predominance of both parasites is common in this region or in the other parts of the world. Studies from Saudi Arabia [74], Ethiopia [75], Sudan [76] and Yemen [77] reported supportive findings. The pooled prevalence rate of *E. histolytica/ dispar* in this meta-analysis was 13.3%, which is consistent with the 14.09% rate reported in Ethiopia [4] and 12.1% in the Philippines [78]. However, the prevalence rate in the present study was lower than that in studies conducted in Malaysia (20.4%) [79], Yemen (16.4) [80] and Tanzania 15% [81] but higher than that in studies in Bangladesh (3.83%) [82] and Thailand (3.7%) [83].

This study showed that 12% of school-aged African children were infected with *Giardia* spp. parasite. Similar infection rate (11.0%) was reported from Brazil [84], and a considerably higher prevalence rate was detected in Nepal (46.8%) [85]. Meanwhile, the infection rate in

Bangladesh (6.01%) [82] and Thailand (4.9%) [83] was lower than the present finding. The variations in prevalence rates of *E. histolytica/dispar* and *Giardia* spp. might be attributed to low sanitation level, contamination of drinking water source, poor hand washing practices and consumption of raw vegetables.

### Strengths and limitations

A key strength of this systematic review and meta-analysis is that it is the first to determine the pooled prevalence estimates of IPPs amongst school-aged children in the entire continent of Africa. Nevertheless, this review has its own limitation. The prevalence data were reported from only 19 of the 54 African countries, and the distribution of eligible studies was uneven across UNSD African regions, publication years and diagnostic methods. Given the limited sensitivity of microscopy to morphologically distinguish between samples infected with *E. histolytica* and those infected with other non-pathogenic *Entamoeba* species, the magnitude of the *E. histolytica* infection might be overestimated because the majority of the included studies used microscopy. Substantial heterogeneity was found across the primary studies, thus generalisations may have limited validity. Overall, the prevalence estimate may not fully represent the continent-wide prevalence of intestinal protozoan infection.

### Conclusion

About 25.8% of school African children had one or more species of intestinal protozoan parasites in their faecal specimens. *E. histolytica/ dispar* and *Giardia* spp. were the most predominant parasites amongst the study participants. This review would be beneficial for understanding the IPPs status amongst African children and provide additional evidence that the burden of these parasites is still alarming. Thus, poverty reduction, improvement of sanitation and hygiene and attention to preventive control measures will be the key to reducing protozoan parasite transmission.

### Supporting information

**S1 Checklist. PRISMA checklist.**
(DOCX)

**S1 Table. Search strategies.**
(DOCX)

**S2 Table. Studies excluded after full text screening.**
(DOCX)

**S3 Table. Quality assessment of the included studies.**
(DOCX)

**S1 Fig.** Subgroup analyses. Prevalence of intestinal protozoan infections among school children in Africa based on children enrolment time (A-C), different regions (D-G), countries (H-Z), diagnostic methods (AA-AC) and species (AD-AG).
(DOCX)

**S2 Fig.** Sensitivity analysis by (A) excluding small studies, (B) excluding low- and moderate-quality studies, (C) excluding studies used non-microscopic diagnostic methods and (D) excluding outlier studies.
(DOCX)

## Acknowledgments

We would like to thank Mr. Mohamad Zarudin Mat Said for his assistance in creating the Map using the QGIS software.

## Author Contributions

**Conceptualization:** Khalid Hajissa, Md Asiful Islam, Zeehaida Mohamed.

**Data curation:** Khalid Hajissa, Md Asiful Islam, Abdoulie M. Sanyang.

**Formal analysis:** Khalid Hajissa, Md Asiful Islam.

**Investigation:** Khalid Hajissa, Md Asiful Islam.

**Methodology:** Khalid Hajissa, Md Asiful Islam, Abdoulie M. Sanyang.

**Resources:** Khalid Hajissa, Md Asiful Islam.

**Software:** Khalid Hajissa, Md Asiful Islam.

**Validation:** Khalid Hajissa, Md Asiful Islam, Zeehaida Mohamed.

**Visualization:** Khalid Hajissa, Md Asiful Islam, Zeehaida Mohamed.

**Writing – original draft:** Khalid Hajissa.

**Writing – review & editing:** Khalid Hajissa, Md Asiful Islam, Zeehaida Mohamed.

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
