## [Decision Letter · Decision Letter 0]

31 Aug 2021

Dear Dr. Issa,

Thank you very much for submitting your manuscript "Prevalence of intestinal protozoan parasites among 29,968 school children in Africa: A systematic review and meta-analysis" for consideration at PLOS Neglected Tropical Diseases. As with all papers reviewed by the journal, your manuscript was reviewed by members of the editorial board and by several independent reviewers. The reviewers appreciated the attention to an important topic. Based on the reviews, we are likely to accept this manuscript for publication, providing that you modify the manuscript according to the review recommendations. 

Sincerely,

Maria Victoria Periago

Deputy Editor

Suzy Campbell

Deputy Editor

Reviewer's Responses to Questions

**Key Review Criteria Required for Acceptance?**

**Methods**

-Are the objectives of the study clearly articulated with a clear testable hypothesis stated?

-Is the study design appropriate to address the stated objectives?

-Is the population clearly described and appropriate for the hypothesis being tested?

-Is the sample size sufficient to ensure adequate power to address the hypothesis being tested?

-Were correct statistical analysis used to support conclusions?

-Are there concerns about ethical or regulatory requirements being met?

Reviewer #1: The objectives of the study are clearly articulated, however is not clear why the authors choose to exclude the helminths from the analysis.

The authors indicated a research question, however a clear testable hypothesis is lacking. Which undermine the scientific impact of the results. Why is important to identify the pooled prevalence of intestinal protozoans in Africa?

Please indicate the inclusion criteria for the meta-analysis; e.g. why small studies (N < 200) were included in the pooled prevalence?

Reviewer #2: (No Response)

Reviewer #3: -Are the objectives of the study clearly articulated with a clear testable hypothesis stated? Yes

-Is the study design appropriate to address the stated objectives? Yes, however only cross-sectional studies were included and surveys in hospitals were excluded. I would suggest not to exclude studies according to study design and to use the design as a stratification criteria in analyses. It is worth to check how many studies were excluded by the authors due to design criteria.

Moreover, the authors classified the results and meta-analyses by year of publication. Instead, it is more relevant to extract and analyze the studies according to the period of data to show a potential evolution of the prevalence of IPPs especially of some species. 

-Is the population clearly described and appropriate for the hypothesis being tested? Yes

-Is the sample size sufficient to ensure adequate power to address the hypothesis being tested? Yes

-Were correct statistical analysis used to support conclusions? Yes, but need clarification for the use of REML method and if any transformations were made for better assessment in meta-analyses (arcsine, logit, Ln..etc)

-Are there concerns about ethical or regulatory requirements being met? Not applicable

**Results**

-Does the analysis presented match the analysis plan?

-Are the results clearly and completely presented?

-Are the figures (Tables, Images) of sufficient quality for clarity?

Reviewer #1: Few recommendations to consider. 

In Fig 2 review the classes descriptions, there are overlapping for the values 45, 60 and 75. For fig 3, would be interesting to add the study location after the study ID. 

In Table 1 DWN stands for Direct Wet Mount? Please add it to the footnote and correct in line 10.

Add in table 3 the 95% CI for the difference of pooled prevalence compared to the main results.

S1 Figure AH is it Escherichia coli or Entanmoeba coli?; AI is it Cryptosporidium parvum or Cryptosporidium spp? 

Line 352, is described that nine types of protozoan were identified, kindly indicate each one of them in the results.

Reviewer #2: (No Response)

Reviewer #3: -Does the analysis presented match the analysis plan? Yes

-Are the results clearly and completely presented? Yes

Recommandations: please add more data to forest plots such as period of data collection or by prevalence. Sorting alphabetically makes the results hard to be interpreted visually.

-Are the figures (Tables, Images) of sufficient quality for clarity? Yes

- Fig 1: please recalculate the totals according to exclusions, they don't match. Please indicate reasons of exclusion of full-texts reviewed and make the list available.

- Table 1 : Please add %, report gender among cases (if available), it is probably more interesting than in the population. Add age of the population, treatment, delay to diagnostic, symptoms..etc.

- Table 2: Please report individual result of prevalence when only one or 2 studies are analysed.

- Fig 3: Pooled estimate is not relevant as there are differences by species and regions..etc. as shown in subgroup analyses. Please sort by %, not alphabetically.

- Fig 4: it is weird that most of the points are outside the funnel and only 3 outliers were identified by the Galbraith plot.

**Conclusions**

-Are the conclusions supported by the data presented?

-Are the limitations of analysis clearly described?

-Do the authors discuss how these data can be helpful to advance our understanding of the topic under study?

-Is public health relevance addressed?

Reviewer #1: (No Response)

Reviewer #2: (No Response)

Reviewer #3: -Are the conclusions supported by the data presented? yes

-Are the limitations of analysis clearly described? yes

-Do the authors discuss how these data can be helpful to advance our understanding of the topic under study? yes

-Is public health relevance addressed? yes

**Editorial and Data Presentation Modifications?**

Reviewer #1: Line 48 replace the dot after Protozoan parasites with a comma.

Line 58 remove "is the first", as all publications should be novel this statement is redundant.

Line 59 "To date...burden of IPPs" is redundant, consider removing.

Line 86 General recommendation, valid for the all manuscript, consider indicating Cryptosporidium spp. and not Cryptosporidium parvum, as there are many species of Cryptosporidium that can cause illness in humans.

Line 87 "Infection by...malaria and schistosomiasis (7)" is not a finding from the reference 7, kindly seek the primary source and update in the manuscript.

Line 88 Replace C. parvum with Cryptosporidium spp. as reference 8 indicates that the most common Cryptosporidium species are hominis and parvum.

Line 115 Replace estiamte with estimate.

Line 128 Is it not supplementary table 2?

Line 141-149 According to the inclusion and exclusion criteria, the following study https://www.ajtmh.org/view/journals/tpmd/81/5/article-p799.xml would be eligible for the analysis, please provide reasons for its exclusion and update the section accordingly.

Line 153 Is it C. parvum or Cryptosporidium spp.? There are other species of Cryptosporidium infecting humans why were the selection restricted to C. parvum?

Line 182 Please indicate what REML stands for.

Line 242 Table 3 and not 2.

Review formation on reference 33.

Table 1 line nº16 Dyab 2016 used modified Zielh-Nelseen to identify Cryptosporidium, which could not indicate the specie, please change in the reported parasite to Cryptosporidium spp.

Line 191-192 "moderate-quality studies (high risk of bias)" should it not be low-quality studies (high risk of bias) as indicated in line 175, please recheck which studies were excluded and update the analysis if required.

Line 282-283 Why were the studies that uses non-microscopic detection method excluded? Aren't those the most sensitive?

Line 285 Were low-and-moderate quality studies removed on the sensitivity analysis or the moderate quality studies?

Reviewer #2: (No Response)

Reviewer #3: Minor revision

Please see suggestions for tables and figures

**Summary and General Comments**

Reviewer #1: Line 285-286 The difference of the overall pooled prevalence (25.8%) compared to the result excluding low-and moderate quality study (16.4%) is significant (p < 0.05). What can explain this finding?

Reviewer #2: (No Response)

Reviewer #3: (No Response)

PLOS authors have the option to publish the peer review history of their article (what does this mean?). If published, this will include your full peer review and any attached files.

Reviewer #1: No

Reviewer #2: Yes: Sabrina John Moyo

Reviewer #3: No

Figure Files:

Data Requirements:

Reproducibility:

References

---

## [Editor Report · Decision Letter 1]

3 Nov 2021

Dear Dr. Hajissa,

We are pleased to inform you that your manuscript 'Prevalence of intestinal protozoan parasites among school children in Africa: A systematic review and meta-analysis' has been provisionally accepted for publication in PLOS Neglected Tropical Diseases.

Best regards,

Maria Victoria Periago

Deputy Editor

Suzy Campbell

Deputy Editor

Thank you for taking in to consideration all the comments from the reviewers and making the necessary modifications. I would only ask you to correct the typo on line 193: "created".

---

## [Editor Report · Acceptance letter]

7 Feb 2022

Dear Dr. Hajissa,

We are delighted to inform you that your manuscript, "Prevalence of intestinal protozoan parasites among school children in Africa: A systematic review and meta-analysis," has been formally accepted for publication in PLOS Neglected Tropical Diseases.

Best regards,

Shaden Kamhawi

co-Editor-in-Chief

Paul Brindley

co-Editor-in-Chief
